# HDC-X: Efficient Medical Data Classification for Embedded Devices

## Abstract

Energy-efficient medical data classification is essential for modern disease screening, particularly in home and field healthcare where embedded devices are prevalent. While deep learning models achieve state-of-the-art accuracy, their substantial energy consumption and reliance on GPUs limit deployment on such platforms. We present HDC-X, a lightweight classification framework designed for low-power devices. HDC-X encodes data into high-dimensional hypervectors, aggregates them into multiple cluster-specific prototypes, and performs classification through similarity search in hyperspace. We evaluate HDC-X across three medical classification tasks; on heart sound classification, HDC-X is $350\times$ more energy-efficient than Bayesian ResNet with less than 1% accuracy difference. Moreover, HDC-X demonstrates exceptional robustness to noise, limited training data, and hardware error, supported by both theoretical analysis and empirical results, highlighting its potential for reliable deployment in real-world settings.

## 1 Introduction

Medical data classification enables automated disease screening without constant involvement of medical professionals (Tan et al., 2024). With the growing adoption of portable devices, health assessments are increasingly carried out in home and field settings where computational resources and power budgets are limited (Rahman & Morshed, 2025). This highlights the need for classification models that can operate effectively on embedded platforms (Khan et al., 2022).

Deep learning models currently dominate medical classification tasks and achieve state-of-the-art accuracy, but their high energy consumption and reliance on GPUs limit deployment on low-power, near-sensor hardware (Chen et al., 2021). In contrast, an optimal medical classifier for embedded devices should: (1) minimize energy consumption, (2) support GPU-free inference, and (3) process data locally to preserve patient privacy.

To address this, we present Hyperdimensional Computing - Next Generation (HDC-X), a lightweight classification framework that extends standard Hyperdimensional Computing (HDC) while significantly improving its performance. HDC-X encodes each sample into a high-dimensional sample hypervector (Sample-HV), aggregates them into a compact set of cluster prototypes (Cluster-HVs), and performs classification by selecting the Cluster-HV with the highest similarity.

Evaluated across three medical classification tasks, HDC-X consumes far less energy than deep learning baselines while achieving notable accuracy gains over standard HDC. On heart sounds classification, HDC-X is $350\times$ more energy-efficient per inference than the state-of-the-art Bayesian ResNet, while providing >10% accuracy improvement over standard HDC (Figure 1). HDC-X also demonstrates exceptional robustness: accuracy drops by only 1.39% under 15% input noise, 1.78% with 40% training data, and 2.84% with 20% parameter corruption (Figure 4). We provide theoretical analysis explaining this robustness, supporting HDC-X's reliability for real-world deployment. Our main contributions are:

- We propose HDC-X, a lightweight classification framework extended from Hyperdimensional Computing (HDC). It achieves significantly higher accuracy than standard HDC and other efficient models while preserving energy efficiency.

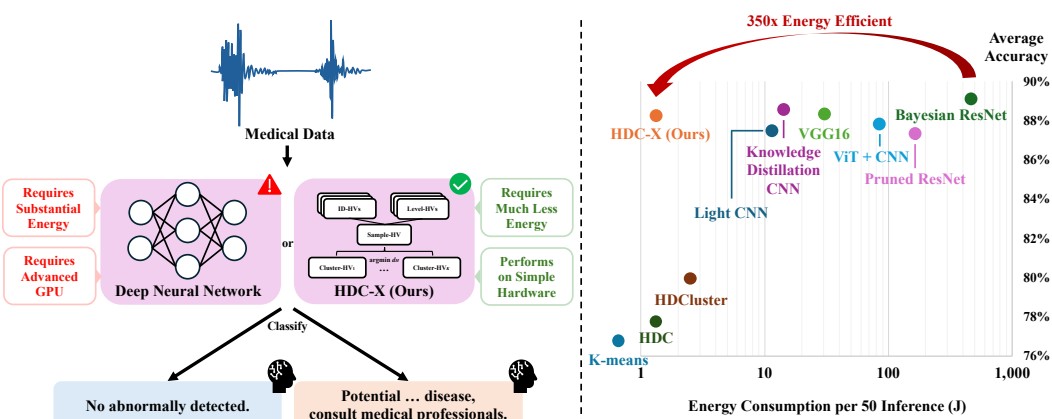

Figure 1: Automated disease screening through medical data classification (left) and HDC-X performance on heart sound classification (right). HDC-X is $350\times$ more energy-efficient than Bayesian ResNet and supports GPU-free inference, highlighting its potential for embedded deployment.

- We demonstrate that HDC-X consumes far less energy than deep learning baselines across three medical tasks. On heart sound classification, it is $350\times$ more energy-efficient than Bayesian ResNet while maintaining comparable accuracy.

- We provide both theoretical and empirical evidence of HDC-X's robustness to input noise and hardware error. To the best of our knowledge, our theorems on binary hypervectors, although some are widely recognized, have not been formally proven in prior HDC research.

This paper is organized as follows. Section 2 reviews the fundamentals of hyperdimensional computing. Section 3 details the proposed HDC-X framework. Section 4 presents experimental results and sensitivity analysis, and Section 5 concludes the paper.

## 2 HYPERDIMENSIONAL COMPUTING (HDC)

Hyperdimensional Computing is a computational paradigm inspired by the information processing mechanisms of the brain (Pentti Kanerva, 2009). Compared with traditional computing that operates on raw numerical data, the human brain processes information via high-dimensional patterns of neural activity (Masse et al., 2009). HDC emulates this approach by projecting input data into high-dimensional representations to efficiently perform cognitive tasks (Thomas et al., 2021).

The versatility of HDC has been demonstrated across a wide array of domains, including image classification (Billmeyer & Parhi, 2021), language identification (Alonso et al., 2021), acoustic signal analysis (Imani et al., 2017), and fault detection (Kleyko et al., 2018). Across these applications, HDC demonstrates superior energy efficiency and robustness compared to deep neural networks.

Specifically, HDC defines a set of elementary operations, like Binding (e.g., point-wise multiplication) and Bundling (e.g., point-wise addition with majority function), on a high-dimensional bipolar vector space $\mathcal{H}^D = \{-1, +1\}^D$ called hyperspace (Pentti Kanerva, 2009). The space is equipped with a distance measure called Hamming distance $d_H : \mathcal{H}^D \times \mathcal{H}^D \to [0, 1]$, which is defined as the ratio of the different bits between two hypervectors (i.e., hyperspace vectors). Please refer to Appendix A.1 for the detailed notations and definitions.

HDC exhibits several ideal properties when its dimensionality $D$ is very large. For example, the Hamming distance of two hypervectors remains unchanged after bound to the same hypervector (Lemma 2); or when randomly selecting two hypervectors, their Hamming distance is almost always around 0.5 (Lemma 3). Based on these properties, HDC can encode a low-dimensional input $s \in \mathbb{R}^d (d \ll D)$ from a continuous feature space into the high-dimensional hyperspace. By transferring the calculation to hyperspace, HDC-based approaches can achieve computational efficiency (Ge & Parhi, 2020), robustness to input noise (Thomas et al., 2021), and higher training data efficiency (Asgarinejad et al., 2020).

## 3 MEDICAL DATA CLASSIFICATION THROUGH HDC-X

We propose Hyperdimensional Computing - Next Generation (HDC-X), an extension of the HDC framework that significantly improves accuracy while preserving energy efficiency. Algorithm 1 and Figure 3 illustrate the overall HDC-X pipeline. In this work, medical samples are first encoded into sample hypervectors (Section 3.1). These sample hypervectors are then clustered into a compact set of cluster prototypes, and classification is performed by similarity search against these prototypes (Section 3.2). An optional retraining stage may further enhance accuracy (Section 3.3). Formal definitions, lemma, and theorems supporting this framework are provided in Appendix A.

### 3.1 ENCODE SAMPLE INTO HYPERVECTOR

HDC-X requires numeric features as input. For heart sound classification, we extract $d = 720$ features using Mel-frequency Cepstral Coefficients (MFCC) (Davis & Mermelstein, 1980) and Discrete Wavelet Transform (DWT) (Mallat, 1989), two widely used frequency-domain representations in audio analysis. For breast cancer classification, we use $d = 30$ features from fine-needle aspirate (FNA) breast mass images, and for EMG classification, we use $d = 8$ features.

Each sample's feature vector $s \in \mathbb{R}^d$ is encoded into a high-dimensional binary hypervector called Sample Hypervector (Sample-HV): $S \in \mathcal{H}^D = \{-1, +1\}^D$, $d \ll D$. The encoder is designed to be continuous so that relative distances between samples are preserved. The first step of encoding divides each feature's value range into $M$ intervals: the middle 96% of values are split into $M$ equal-width intervals, while the top and bottom 2% are directly mapped to the first and last intervals (Figure 2). This is formalized by function $l$ in Definition 5.

Each interval in Figure 2 is represented by a predefined Level Hypervector (Level-HV), denoted as $L^{(m)} \in \mathcal{H}^D$, $m \in \{1, 2, \ldots, M\}$. The Level-HVs are generated in a way that neighboring Level-HVs have a low Hamming distance (i.e. low percentage of different bits): starting from a randomly generated $L^{(1)}$, each subsequent Level-HV is formed by randomly flipping $D/(M-1)$ bits from the previous

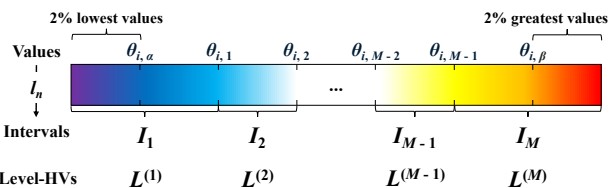

Figure 2: Divide feature $n$'s value range into $M$ intervals.

vector, with each bit flipped only once across the sequence (Definition 4). This ensures that the Hamming distance between any two Level-HVs satisfies:

$$d_H(L^{(i)}, L^{(j)}) = \frac{|i - j|}{M - 1}. \tag{1}$$

The Level-HVs are shared across all features. To distinguish identical Level-HVs under different features, each feature is assigned a randomly sampled Identity Hypervector (ID-HV), denoted as $ID^{(n)} \in \mathcal{H}^D$, $n \in \{1, 2, \ldots, d\}$. A value in interval $I_m$ under feature $n$ would be represented as $ID^{(n)} \otimes L^{(m)}$, where $\otimes$ denotes the binding operation implemented by point-wise multiplication (Definition 2). Since ID-HVs are randomly sampled, their pairwise Hamming distance is approximately 0.5 (Lemma 3), ensuring feature-wise independence: Hamming distance between two Level-HVs remains unchanged after binding to the same ID-HV (Lemma 2), while those bound to different ID-HVs become effectively unrelated.

Finally, with the predefined Level-HVs and ID-HVs, the encoder maps each medical sample to a Sample-HV by bundling the representations of each feature:

$$S^{(i)} = [\sum_{n=1}^{d} ID^{(n)} \otimes L^{(l_n(s_n^{(i)}))}], \tag{2}$$

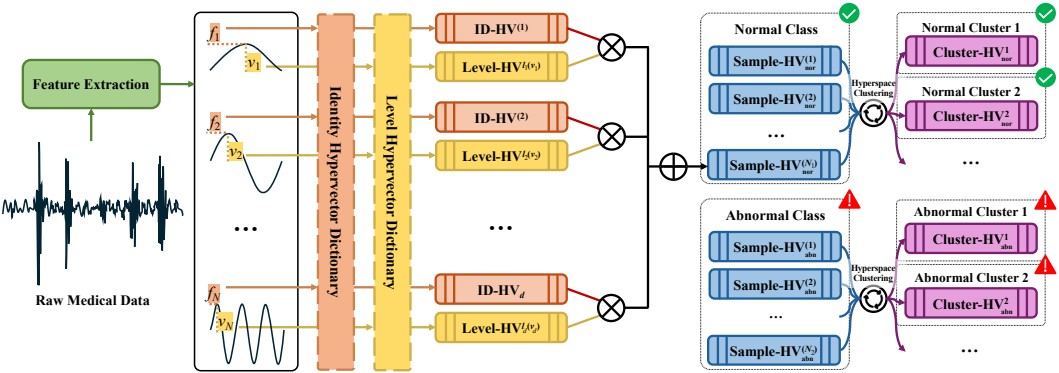

Figure 3: Medical data classification through HDC-X: Training samples are encoded into sample hypervectors (Sample-HVs) and aggregated into a compact set of cluster prototypes (Cluster-HVs); new samples are classified by selecting the Cluster-HV with highest similarity. The figure illustrates a binary classification example, though HDC-X is not limited to binary tasks.

where $s_n^{(i)}$ denotes the value of the $n^{th}$ feature in sample $s^{(i)}$, and $[\cdot]$ is the element-wise majority function: it outputs $+1$ for positive sum, $-1$ for negative sum, and equally random samples from $\{-1, +1\}$ for zero sum (Definition 2).

We demonstrate the inherent robustness of this encoding to input noise through Theorem 1. When noise is applied to a feature vector $s^{(1)}$, producing a perturbed version $s^{(2)}$, the theorem shows that as long as the noise is bounded by a relative ratio $\delta$, the Hamming distance between the corresponding Sample-HVs $S^{(1)}$ and $S^{(2)}$ has an upper bound. This upper bound, expressed as $g(\delta)$, remains quantitatively small even under moderate noise. The formal proof and definition of $g(\delta)$ are provided in Proof 1

**Theorem 1** (Robustness to Input Noise). *Let $s^{(1)}, s^{(2)} \in \mathcal{S} \subseteq \mathbb{R}^d$ be two feature vectors. $S = f_{\mathcal{ID},\mathcal{L},\Theta}(s)$ denotes the hypervector encoding defined in Equation 2. Suppose that for all indices of the features $n \in \{1, \ldots, d\}$,*

$$\frac{\left| x_n^{(1)} - x_n^{(2)} \right|}{\Delta_i} \leq \delta, \tag{3}$$

*where $\Delta_i$ is the difference between the upper and lower bounds of the $i^{th}$ feature value, and $\delta \in [0, 1]$ denotes the maximum distance ratio of the corresponding features. Then, with a sufficiently large D, the expected upper-bound Hamming distance between $S^{(1)} = f_{\mathcal{ID},\mathcal{L},\Theta}(s^{(1)})$ and $S^{(2)} = f_{\mathcal{ID},\mathcal{L},\Theta}(s^{(2)})$ converges to a monotonically increasing function g with parameter $\delta$:*

$$\mathbb{E}\left[ \sup \left( d_H \left( S^{(1)}, S^{(2)} \right) \right) \right] \to g(\delta). \tag{4}$$

*This implies that the Hamming distance between a sample and its noisy variant is almost surely bounded by $g(\delta)$.*

## 3.2 CLASS-WISE HYPERSPACE CLUSTERING

Medical data often exhibit substantial intra-class variability beyond simple class labels. For example, heart sounds sharing the same 'abnormal' label can differ due to the type and stage of cardiac disease, the stethoscope used, and the recording site on the body (Zipes et al., 2019). Such heterogeneity challenges the standard HDC pipeline, which rely on a single prototype to represent all samples within one class.

To address this, HDC-X introduces class-wise hyperspace clustering. The clustering process is inspired by K-means, but performed in hyperspace on Sample-HVs and applied independently within each class. Unlike K-means which computes arithmetic means, HDC-X uses the bundling operation (Definition 2) to form cluster prototypes (Cluster-HVs). Leveraging hypervectors rather than raw numeric features provides the robustness formalized in Theorem 1.

---

**Algorithm 1** HDC-X: Hyperdimensional Computing - Next Generation (Without Retraining)

---
1: $\{ID^{(n)}\}_{n=1}^{d} \leftarrow$ identity_hypervectors$(d)$     ▷ Generate ID-HV Dictionary.
2: $\{L^{(m)}\}_{m=1}^{M} \leftarrow$ level_hypervectors$(M)$     ▷ Generate Level-HV Dictionary.
3: **for** $s^{(i)} \in \mathcal{S}$ **in** training set **do**
4:   $S^{(i)} \leftarrow [\sum_{n=1}^{d} ID^{(n)} \otimes L^{(l_n(s_n^{(i)}))}]$    ▷ Encode sample $s^{(i)}$ into Sample-HV $S^{(i)}$.
5: **end for**
6: **for** $j = 1$ **to** number of classes $J$ **do**
7:   **for** $S_j^{(i)} \in$ class $j$ **do**
8:    init_cluster_idx $\leftarrow$ randint$(K)$     ▷ Random assign $S_j^{(i)}$ to one of $K$ clusters.
9:    $\mathfrak{C}_j^{\text{init\_cluster\_idx}} \leftarrow \mathfrak{C}_j^{\text{init\_cluster\_idx}} \cup \{S_j^{(i)}\}$
10:   **end for**
11:   **for** $t = 1$ **to** number of clustering iterations $T$ **do**
12:    **for** $k = 1$ **to** number of clusters $K$ **do**
13:     $C_j^k \leftarrow [\sum_{S_j^{(i)} \in \mathfrak{C}_j^k} S_j^{(i)}]$    ▷ Generate Cluster-HV $C_j^k$ to represent cluster $\mathfrak{C}_j^k$.
14:    **end for**
15:    **for** $S_j^{(i)} \in$ class $j$ **do**
16:     cluster_idx $\leftarrow \text{argmin}_{k=1}^{K} d_H(S_j^{(i)}, C_j^k)$    ▷ Reassign $S_j^{(i)}$ to closest cluster.
17:     $\mathfrak{C}_j^{\text{cluster\_idx}} \leftarrow \mathfrak{C}_j^{\text{cluster\_idx}} \cup \{S_j^{(i)}\}$
18:    **end for**
19:   **end for**
20: **end for**
21: (class_idx, cluster_idx) $\leftarrow \text{argmin}_{(j,k) \in [J] \times [K]} d_H(S_j^{\text{new}}, C_j^k)$    ▷ Classify new sample.
22: **return** class_idx

---

Specifically, for each class $j$, clustering begins by randomly assigning its Sample-HVs $\{S_j^{(i)}\}$ to $K$ clusters $\{\mathfrak{C}_j^k\}_{k=1}^{K}$. Each Cluster-HV $C_j^k \in \mathcal{H}^D$ is computed by bundling the Sample-HVs in that cluster:

$$C_j^k = [\sum_{S_j^{(i)} \in \mathfrak{C}_j^k} S_j^{(i)}]. \tag{5}$$

Next, the Sample-HVs are reassigned to the cluster whose prototype has the lowest Hamming distance:

$$S_j^{(i)} \rightarrow \underset{k \in [K]}{\text{argmin}} \, d_H(S_j^{(i)}, C_j^k), \qquad [K] := \{1, 2, \ldots, K\}. \tag{6}$$

This bundling and reassigning process is repeated for $T$ iterations until convergence, yielding the final set of Cluster-HVs $\{C_j^k\}_{k=1}^{K}$ for class $j$. After clustering all $J$ classes, HDC-X may classify an unseen new sample by encoding it as Sample-HV $S^{\text{new}}$ and assigning it to the Cluster-HV with the highest similarity (i.e., lowest Hamming distance):

$$S^{\text{new}} \rightarrow \underset{(j,k) \in [J] \times [K]}{\text{argmin}} \, d_H(S^{\text{new}}, C_j^k). \tag{7}$$

The sample is then classified according to the class label of the selected cluster.

Theorem 2 provides theoretical insight into our clustering-by-bundling method in Equation 5 (Proof 2). It shows that the Hamming distance between the Cluster-HV $C$ and any Sample-HV that constitutes the cluster is almost always less than the typical distance of $d_H = 0.5$ between two random hypervectors. This supports that the Cluster-HV can sufficiently represent all samples in the cluster. Moreover, since Sample-HVs within a cluster are typically more similar to one another than two randomly drawn hypervectors, the Cluster-HV is expected to preserve even stronger relationships in practice than those guaranteed under the random hypervector assumption in the theorem.

**Theorem 2** (Distance Between Cluster Prototype and Constituents). *Let $S^{(1)}, S^{(2)}, \ldots, S^{(N)} \in \mathcal{H}^D$ be independently sampled random hypervectors. Define their bundling sum as $C = [S^{(1)} + S^{(2)} + \cdots + S^{(N)}]$. As $D \to \infty$, for any random hypervector $S^* \in \mathcal{H}^D$, index $j \in \{1, \ldots, N\}$, the Hamming distance between $C$ and any component $S^{(n)}$ satisfies*

$$P\left(d_H(C, S^{(n)}) < d_H(C, S^*)\right) \to 1. \tag{8}$$

### 3.3 Retrain Cluster Prototypes

To further improve accuracy, HDC-X optionally applies a retraining procedure that adjusts Cluster-HVs based on misclassified training samples. The retraining stage operates exclusively on the training set with no reference to test data.

Recall that each Cluster-HV $C_j^k$ is generated by bundling the Sample-HVs assigned to cluster $\mathfrak{C}_j^k$, i.e., $C_j^k = [\sum_{S_j^{(i)} \in \mathfrak{C}_j^k} S_j^{(i)}]$. Let $S^{\text{err}}$ be a misclassified Sample-HV from the training set. Suppose $S^{\text{err}} \in \mathfrak{C}_{j_1}^{k_1}$, but its nearest Cluster-HV in Hamming distance is a different Cluster-HV $C_{j_2}^{k_2}$. This implies that the Cluster-HV $C_{j_1}^{k_1}$, constructed from cluster $\mathfrak{C}_{j_1}^{k_1}$, does not adequately represent $S^{\text{err}}$.

To correct this, HDC-X performs two adjustments: (1) Subtracts $S^{\text{err}}$ from the incorrect Cluster-HV $C_{j_2}^{k_2}$ and (2) Re-bundles $S^{\text{err}}$ onto the correct Cluster-HV $C_{j_1}^{k_1}$. We generalize this operation across all misclassified training samples. Let $\mathcal{E}_j^{k,\text{out}}$ denote Sample-HVs outside cluster $\mathfrak{C}_j^k$ that incorrectly match closest to $C_j^k$; while $\mathcal{E}_j^{k,\text{in}}$ denote the Sample-HVs within $\mathfrak{C}_k^j$ but are closer to a Cluster-HV of a different cluster. The retrained Cluster-HV representing cluster $\mathfrak{C}_k^j$ is computed as

$$C_j^{k\prime} = [\sum_{S_j^{(i)} \in \mathfrak{C}_j^k} S_j^{(i)} - \sum_{S_j^{(i)} \in \mathcal{E}_j^{k,\text{out}}} S_j^{(i)} + \sum_{S_j^{(i)} \in \mathcal{E}_j^{k,\text{in}}} S_j^{(i)}]. \tag{9}$$

## 4 Results and Discussions

### 4.1 Datasets

**PhysioNet/CinC Challenge 2016** (Clifford et al., 2016) provides a collection of 3,153 heart sound recordings sourced from six distinct databases. These recordings were collected by multiple research teams across various countries using different equipment and methodologies, often under noisy conditions. Each recording is labeled as either 'normal' or 'abnormal'.

**Wisconsin Breast Cancer** (William Wolberg et al., 1995) is a widely used benchmark for breast cancer diagnosis. It contains real-valued features extracted from digitized images of fine-needle aspirates (FNA) of breast masses, with each sample labeled as either 'benign' or 'malignant'.

**Cerqueira sEMG Muscle Fatigue** (Cerqueira et al., 2024) offers surface electromyography (sEMG) recordings using a Delsys Trigno system at a sampling frequency of 1259Hz. Each sample is annotated into three fatigue levels: relax, mild fatigue, and fatigue.

### 4.2 Experiment Setup

**Platform:** Large deep learning baselines were trained and evaluated on an NVIDIA GeForce RTX 4090 GPU (24GB VRAM). HDC-X, other one-shot models, and efficient neural networks were trained and evaluated on an Intel Xeon Gold 6430 CPU (120GB RAM). All experiments were conducted on Ubuntu 22.04.1 LTS (Linux kernel 5.15.0-97-generic).

**Energy Measurement:** We define *energy* as device-level energy consumption (joules). Training and inference energy were computed by integrating measured power over time.

Table 1: Performance and Energy Comparison of HDC-X and Baseline Models

| Models | 10-Folds Accuracy (%) | Energy (Train, J) | Energy (1000 Inferences, J) |
|---|---|---|---|
| **PhysioNet Challenge 2016:** | | | |
| Bayesian ResNet (H. Krones et al., 2022) | 89.105 ± 1.543 | 142997 ± 5465 | 9455 ± 899 |
| Knowledge Distillation CNN (Song et al., 2023) | 88.580 ± 2.186 | 32808 ± 2582 | 289 ± 116 |
| VGG16 (Shuvo et al., 2023) | 88.271 ± 1.718 | 165840 ± 5443 | 605 ± 131 |
| ViT + CNN (Han & Shaout, 2025) | 87.808 ± 1.996 | 889920 ± 36748 | 1661 ± 209 |
| Light CNN (Li et al., 2021) | 87.408 ± 1.497 | 10164 ± 1204 | 227 ± 88 |
| Pruned Bayesian ResNet (H. Krones et al., 2022) | 87.190 ± 2.508 | 142997 ± 5465 | 3307 ± 214 |
| Class-Wise K-Means | 76.759 ± 4.937 | 9 ± 2 | *very low* |
| HDCluster (Imani et al., 2019) | 79.877 ± 2.589 | 318 ± 3 | 50 ± 2 |
| HDC | 77.840 ± 2.419 | 135 ± 10 | 26 ± 4 |
| **HDC-X (Ours)** | **88.180 ± 1.746** | **246 ± 6** | **27 ± 3** |
| **Wisconsin Breast Cancer:** | | | |
| DNN (Zheng, 2024) | 96.842 ± 2.579 | 698.626 ± 82.403 | 11.842 ± 1.470 |
| GRU (Jony & Arnob, 2024) | 96.668 ± 1.994 | 2710.461 ± 109.069 | 97.283 ± 2.408 |
| CNN (Jony & Arnob, 2024) | 95.439 ± 2.956 | 312.416 ± 7.874 | 12.458 ± 0.368 |
| Class-Wise K-Means | 93.595 ± 3.106 | 3.045 ± 0.379 | 0.070 ± 0.007 |
| HDC | 94.382 ± 0.039 | 2.489 ± 0.174 | 0.717 ± 0.021 |
| **HDC-X (Ours)** | **96.314 ± 0.027** | **3.801 ± 0.340** | **0.823 ± 0.015** |
| **Cerqueira sEMG Muscle Fatigue:** | | | |
| GRU (Aviles et al., 2024) | 91.955 ± 3.084 | 874.511 ± 42.378 | 34.018 ± 1.013 |
| LSTM (Aviles et al., 2024) | 91.710 ± 2.705 | 135.828 ± 7.453 | 32.416 ± 1.483 |
| CNN (Moniri et al., 2021) | 91.367 ± 3.626 | 250.685 ± 21.217 | 10.426 ± 0.884 |
| Class-Wise K-Means | 88.596 ± 5.135 | 3.367 ± 0.247 | 0.076 ± 0.010 |
| HDC (Moin et al., 2018) | 84.984 ± 0.043 | 2.634 ± 0.169 | 0.445 ± 0.025 |
| **HDC-X (Ours)** | **91.592 ± 2.927** | **2.813 ± 0.205** | **0.422 ± 0.029** |

## 4.3 COMPARISON WITH BASELINE MODELS

Table 1 summarizes the performance of HDC-X against baseline models across three medical datasets. HDC-X achieves substantially higher energy efficiency than deep neural networks and outperforms other efficient methods in terms of accuracy. For instance, on the heart sounds classification task, HDC-X is $580\times$ more energy-efficient in training and $350\times$ in inference compared to the best-performing Bayesian ResNet (H. Krones et al., 2022), while also delivering a 10% accuracy improvement over standard HDC.

## 4.4 IMPACT OF HDC-X HYPERPARAMETERS

As discussed in Section 3, a sufficiently large dimensionality $D$ is critical for HDC-X to maintain robustness against input noise (Theorem 1) and to ensure that Cluster-HVs accurately capture the aggregate features of their clusters (Theorem 2). Likewise, an adequate number of clusters $K$ are essential for forming stable hyperspace clusters. A few retrain epochs can further fine-tune Cluster-HVs to better align with the Sample-HVs assigned to each cluster.

However, excessively large values of these hyperparameters introduce unnecessary computational overhead, reducing the energy efficiency HDC-X is designed to achieve. Our experiments also indicate that allocating too many clusters or performing excessive retraining can lead to overfitting. Figure 4 (a-c) illustrates HDC-X's performance on heart sound classification across different hyperparameter settings.

## 4.5 ROBUSTNESS FOR REAL-WORLD DEPLOYMENT

### 4.5.1 RESILIENCE TO INPUT NOISE

Real-world medical signals often contain persistent environmental noise, which can impair classification performance (Clifford et al., 2016). The inherent robustness of HDC-X's encoding mechanism, as supported by Theorem 1, allows it to maintain high accuracy even with noisy inputs. As shown in Figure 4(d), HDC-X experiences only 1.39% drop in accuracy under 15% input noise, demonstrating its reliability in noisy settings.

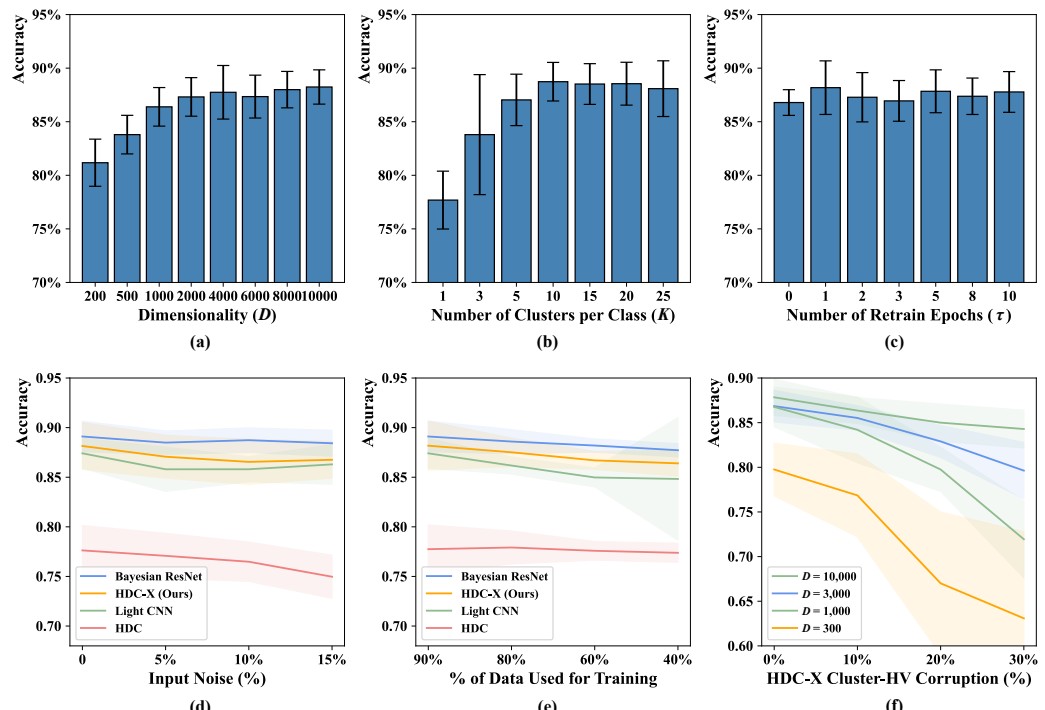

Figure 4: HDC-X sensitivity to hyperparameters, input noise, limited training data, and hardware errors on PhysioNet 2016.

### 4.5.2 RESILIENCE TO LIMITED TRAINING DATA

Many medical datasets are limited in size due to privacy constraints. As shown in Figure 4(e), HDC-X remains robust under reduced training data, with only a 1.78% drop in accuracy when trained on 40% of the PhysioNet 2016 dataset.

### 4.5.3 RESILIENCE TO HARDWARE ERRORS

Neural architectures are known for their fault tolerance through redundant representations, unlike traditional binary systems where single-bit failures are critical (Pentti Kanerva, 2009). Similarly, brain-inspired hyperdimensional encodings offer inherent robustness to hardware malfunctions by distributing information across high-dimensional vectors (Kanerva et al., 2000).

To evaluate this robustness in HDC-X, we conducted a perturbation analysis by randomly flipping elements (+1 to -1 and vice versa) in all stored Cluster-HVs to simulate hardware instability. Theorem 3 demonstrates that, with sufficient dimensionality, flipping up to 50% of the elements has minimal impact on classification accuracy (Proof 3). The theorem is supported empirically: as shown in Figure 4(f), with $D = 10,000$, flipping 20% of elements results in only a 2.84% drop in accuracy.

**Theorem 3** (Robustness to Hardware Error). *Assume we have a Sample-HV $S$ and two Cluster-HVs $C_1$ and $C_2$, whose initial Hamming distances satisfy:*

$$d_H(S, C_1) = d_1 \quad and \quad d_H(S, C_2) = d_2, \tag{10}$$

*where $d_2 - d_1 = \epsilon > 0$.*

*We randomly flip a proportion $p$ ($p < 0.5$) of the bits in both $C_1$ and $C_2$, yielding two new hypervectors $C_1'$ and $C_2'$. As $D \to \infty$, we have*

$$P\big(d_H(S, C_1') < d_H(S, C_2')\big) \to 1 \tag{11}$$

### 4.6 CONCEPTUAL HARDWARE FRAMEWORK

The current implementation of HDC-X is in Python, which does not fully exploit the hardware-level efficiency of binary hypervectors enabled by single-bit operations and parallel computing. Prior work on HDC hardware has shown that fundamental operations such as binding and bundling can be implemented with extremely lightweight digital logic: binding reduces to bitwise XOR, bundling to majority voting circuits, and similarity search to parallelized Hamming distance computation (Imani et al., 2017). These properties allow HDC systems to operate with low energy consumption and high throughput compared to conventional floating-point ML models.

Building on these insights, we outline a conceptual hardware framework specifically tailored for HDC-X (Figure 5). To

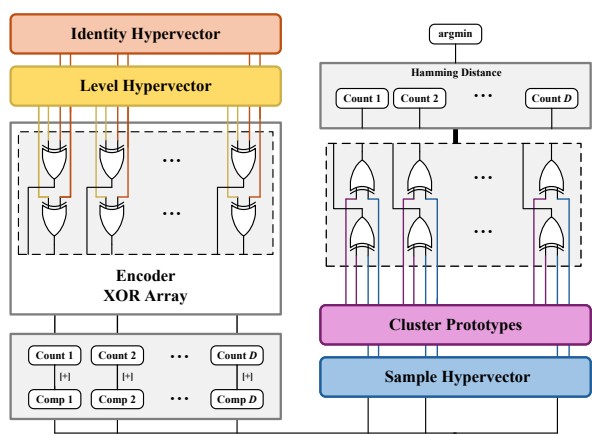

Figure 5: Conceptual hardware framework for HDC-X.

validate its feasibility, we prototyped the framework on a Xilinx FPGA platform and are able to achieve the accuracy reported in Table 1. We plan to further refine and evaluate this hardware-oriented design in future work.

## 5 CONCLUSION

This paper introduces HDC-X, an energy-efficient classification framework extended from Hyperdimensional Computing (HDC). HDC-X demonstrates significant advantage in medical data classification: on heart sound classification, it is $350\times$ more efficient than Bayesian ResNet, and provide $> 10\%$ accuracy improvement over standard HDC.

We evaluated HDC-X's robustness both theoretically and empirically. The model maintains high accuracy under challenging conditions: showing only a 1.39% drop under 15% input noise, 1.78% when trained on just 40% of the dataset, and 2.84% when 20% of its learned parameters are corrupted. These results highlight HDC-X's reliability on real-world medical applications.

From a societal perspective, HDC-X can help expand access to medical screening in underserved settings by enabling assessments on low-cost, GPU-free devices. However, overreliance on the screening tool without clinical oversight could lead to misdiagnoses; therefore, the system should be designed to recommend follow-up with a medical professional when abnormalities are detected.

### REPRODUCIBILITY STATEMENT

We are committed to ensuring the reproducibility of our results. To this end, we have released the complete HDC-X codebase along with a detailed installation guide in the Supplementary Materials. The repository contains all datasets used in our experiments as well as the exact hyperparameter configurations adopted for training and evaluation. This enables researchers to fully replicate our experiments and verify the reported results.

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

# A  THEORY

## A.1  NOTATIONS AND DEFINITIONS

**Definition 1** (Hyperspace). *The hyperspace $\mathcal{H}^D$ is the set of all bipolar hypervectors of dimension $D$, where each element takes a value from $\{+1, -1\}$. Formally,*

$$\mathcal{H}^D := \{-1, +1\}^D = \{A \in \mathbb{R}^D \mid a_i \in \{+1, -1\}, \; \forall i \in \{1, 2, \ldots, D\}\} \tag{12}$$

*where $a_i$ denotes the $i$-th component of the hypervector $A$.*

*Unless otherwise specified, for a random hypervector $A \in \mathcal{H}^D$, each component $a_i$ is drawn independently and identically distributed (i.i.d.) from a binary distribution over $\{-1, +1\}$ with equal probability:*

$$\mathbb{P}(a_i = +1) = \mathbb{P}(a_i = -1) = 0.5, \quad \forall i \in \{1, \ldots, d\}. \tag{13}$$

**Definition 2** (Elementary Functions). *Given any hypervectors $A, B \in \mathcal{H}^D$, we define the following elementary functions:*

- *The number of $+1$ elements in $A$:*

$$pos(A) := \sum_{i=1}^{D} \mathbb{I}(a_i = +1) \tag{14}$$

- *The number of $-1$ elements in $A$:*

$$neg(A) := \sum_{i=1}^{D} \mathbb{I}(a_i = -1) \tag{15}$$

- *Binding of $A$ and $B$:*

$$A \otimes B := A \times B \tag{16}$$

- *Bundling of $A$ and $B$:*

$$A \oplus B := [A + B] \tag{17}$$

*where $\mathbb{I}(\cdot)$ denotes the indicator function that returns 1 if the condition is true and 0 otherwise, $\times$ is an element-wise multiplication, $+$ is an element-wise addition, and $[\cdot]$ is the element-wise majority function that outputs $-1$ for negative sum, $+1$ for positive sum, and equally random drawn from $\{-1, +1\}$ for zero sum.*

**Definition 3** (Hamming Distance). *Given two hypervectors $A, B \in \mathcal{H}^D$, the Hamming distance between them is defined as the ratio of positions where the corresponding components differ. Formally,*

$$d_H : \mathcal{H}^D \times \mathcal{H}^D \to [0, 1], \ \ d_H(A, B) = \frac{1}{D} \sum_{i=1}^{D} \mathbb{I}(a_i \neq b_i) = \frac{1}{D} neg(A \otimes B). \tag{18}$$

**Definition 4** (Level Set). *We define a set of M hypervectors $\mathcal{L} = \{L^{(j)}\}_{j \in \{1,2,\cdots,M\}} \subset \mathcal{H}^D$ as a level set if it can be generated by:*

- *Randomly sample a base hypervector $L^{(1)} \in \mathcal{H}^D$*

- *Initialize an empty set $\mathcal{B} = \emptyset$ to record flipped bit positions.*

- *For each $i \in \{2, 3, \dots, M\}$:*

  - *Randomly select $\frac{D}{M-1}$ positions from $\{1, \dots, D\} \setminus \mathcal{B}$ (i.e., bits not yet flipped). To avoid unnecessary complexity, we assume that $M - 1$ divides $D$ exactly here, i.e., $\frac{D}{M-1} \in \mathbb{N}$.*
  - *Flip the selected bits in $L^{(i-1)}$ to obtain $L^{(i)}$;*
  - *Update $\mathcal{B}$ to include the newly flipped bit positions.*

*In the constructed level set $\mathcal{L}$, the Hamming distance between any two levels $L^{(i)}$ and $L^{(j)}$ satisfies*

$$d_H(L^{(i)}, L^{(j)}) = \frac{|i - j|}{M - 1}.$$

**Definition 5** (Mapping with Levels). *Let $\mathcal{X} \subseteq \mathbb{R}^d$ be a continuous feature vector space. Define independently sampled $d$ random hypervectors $\mathcal{ID} = \{ID^{(1)}, ID^{(2)}, \dots, ID^{(d)}\} \subset \mathcal{H}^D$ and a random level set $\mathcal{L} = \{L^{(1)}, L^{(2)}, \dots, L^{(M)}\} \subset \mathcal{H}^D$. Next, for each feature dimension $n \in \{1, \dots, d\}$, define $M - 1$ real-valued thresholds $\theta_{n,1} < \theta_{n,2} < \cdots < \theta_{n,M-1}$, partitioning $\mathbb{R}$ into $M$ intervals:*

$$I_0 = (-\infty, \theta_{n,1}), \quad I_1 = [\theta_{i,1}, \theta_{n,2}), \quad \dots, \quad I_{M-1} = [\theta_{n,M-1}, +\infty).$$

*Based on the intervals defined by $\Theta = \{\theta_{n,m}\}_{n \in \{1,\dots,d\}, m \in \{1,\dots,M\}}$, we select the unique $m = l_n(x_i[n])$ for $x_i[n] \in \mathbb{R}$ that $x_i[n] \in I_m$.*

*In this task, the mapping function for feature $n$ is defined by*

$$l_n(x) = \begin{cases} 1 & x \in (-\infty, \theta_{n,\alpha}) \\ \lfloor \frac{x - \theta_{n,\alpha}}{\theta_{n,\beta} - \theta_{n,\alpha}} \times M + 1 \rfloor & x \in [\theta_{n,\alpha}, \theta_{n,\beta}) \\ M & x \in [\theta_{n,\beta}, \infty) \end{cases},$$

*where $\theta_{n,\alpha}$ and $\theta_{n,\beta}$ denote the 2% and 98% quantiles of values in feature $n$, respectively.*

## A.2 LEMMAS AND PROPOSITIONS

**Lemma 1** (Hamming Distance between product and multiplier). *Given hypervectors $A, B \in \mathcal{H}^D$, the Hamming distance $d_H(A, A * B)$ depends only on $B$. Specifically,*

$$d_H(A, A \times B) = \frac{1}{D}neg(B). \tag{19}$$

*Proof.* By the definition of Binding, the $i$-th element of $A * B$ is $a_i * b_i$, which behaves as:

$$a_i * b_i = \begin{cases} a_i, & \text{if } b_i = +1, \\ -a_i, & \text{if } b_i = -1. \end{cases} \quad \text{or } a_i \neq (a_i * b_i) \iff b_i = -1$$

Therefore,

$$d_H(A, A * B) = \frac{1}{D} \sum_{i=1}^{D} \mathbb{I}(a_i \neq (a_i * b_i)) = \frac{1}{D} \sum_{i=1}^{D} \mathbb{I}(b_i = -1) = \frac{1}{D}\text{neg}(B).$$

$\square$

**Lemma 2** (Hamming Distance Preservation under Multiplication). *Given hypervectors $A, B, C \in \mathcal{H}^D$, we have*

$$d_H(A * B, B * C) = d_H(A, C). \tag{20}$$

*Proof.* By the definition of Binding, the $i$-th elements of $A * B$ and $B * C$ are $a_i * b_i$ and $b_i * c_i$.

Since $b_i \neq 0$, we have $a_i * b_i \neq b_i * c_i \iff a_i \neq c_i$. Therefore,

$$d_H(A * B, B * C) = \frac{1}{D} \sum_{i=1}^{D} \mathbb{I}(a_i * b_i \neq b_i * c_i) = \frac{1}{D} \sum_{i=1}^{D} \mathbb{I}(a_i \neq c_i) = d_H(A, C).$$

$\square$

**Lemma 3** (Hamming distance between two Random Hypervectors). *Let $A, B \in \mathcal{H}^D$ be two random hypervectors, Then, for any $\epsilon > 0$,*

$$\lim_{D \to \infty} \mathbb{P}\left(|d_H(A, B) - 0.5| > \epsilon\right) = 0. \tag{21}$$

*Proof.* Define random variables $X_i := \mathbb{I}(a_i \neq b_i)$, where $X_i = 1$ if $a_i \neq b_i$ and $X_i = 0$ otherwise. By definition,

$$d_H(A, B) = \frac{1}{D} \sum_{i=1}^{D} X_i.$$

Since $A$ and $B$ are random hypervectors, their components $a_i$ and $b_i$ are drawn i.i.d. from a binary distribution over $\{-1, +1\}$ with equal probability. Thererfore,

$$\mathbb{P}(a_i \neq b_i) = 0.5, \quad \mathbb{P}(a_i = b_i) = 0.5,$$

indicating that each $X_i$ is an independent Bernoulli random variable with $\mathbb{E}[X_i] = 0.5$. Furthermore, we know that the average of i.i.d. sequence $\{X_i\}$, $\frac{1}{D} \sum_{i=1}^{D} X_i$ follows a scaled Binomial distribution:

$$d_H(A, B) = \frac{1}{D} \sum_{i=1}^{D} X_i \sim \frac{1}{D}\text{Binomial}(D, \frac{1}{2}).$$

Applying the Weak Law of Large Numbers to the i.i.d. sequence $\{X_i\}$, for any $\epsilon > 0$,

$$\lim_{D \to \infty} \mathbb{P}\left(\left|\frac{1}{D} \sum_{i=1}^{D} X_i - 0.5\right| > \epsilon\right) = 0.$$

Thus,

$$\lim_{D\to\infty} \mathbb{P}\left(|d_H(A, B) - 0.5| > \epsilon\right) = 0.$$

□

**Lemma 4.** *Let $A, B \in \mathcal{H}^D$ be two random hypervectors, Then, for any $\epsilon > 0$,*

$$\lim_{D\to\infty} \mathbb{P}\left(|d_H(A, A * B) - 0.5| > \epsilon\right) = 0.$$

*Proof.* From the Lemma 1, we know that

$$d_H(A, A * B) = \frac{1}{D}\sum_{i=1}^{D} \mathbb{I}(b_i = -1) = \frac{\text{neg}(B)}{D}.$$

Since $B$ is a random hypervector, each component $b_i$ is drawn i.i.d. from a binary distribution over $\{-1, +1\}$ with equal probability. Again, by the Weak Law of Large Numbers,

$$\lim_{D\to\infty} \mathbb{P}\left(\left|\frac{\text{neg}(B)}{D} - 0.5\right| > \epsilon\right) = 0,$$

thus leading to

$$\lim_{D\to\infty} \mathbb{P}\left(|d_H(A, A * B) - 0.5| > \epsilon\right) = 0.$$

□

**Proposition 1** (Hamming Distance with sufficiently large $D$). *Let two hypervectors $A, B \in \mathcal{H}^D$ that satisfy*

$$d_H(A, B) = \delta,$$

*and the random variable $Z_i$ indicates the situation of the $i^{th}$ bit*

$$Z_i = \mathbb{I}\left(a_i \neq b_i\right).$$

*With sufficiently large $D$, the $Z_i$ can be approximately viewed as an i.i.d. Bernoulli distribution*

$$Z_i \sim_{i.i.d.} Bernoulli(\delta), \forall i \in \{1, 2, \ldots, D\}.$$

*Proof.* With the definition of Hamming distance, we have

$$\frac{1}{D}\sum_{i=1}^{D} Z_i = d_H(A, B) = \delta.$$

Observe that this is an empirical mean of $\{Z_i\}$ over $D$ bits. Notice that the $Z_i$ are NOT independent — their sum is fixed to be exactly $\delta D$. However, we make the following observations and assumptions to justify such a approximation:

- With sufficiently large D, the possibility of the Hamming distance of A and B completely falls into an arbitrarily small interval around the $\delta$ is almost 1

$$\mathbb{P}\left(\left|\frac{1}{D}\sum_{i=1}^{D} Z_i - d\right| > \epsilon\right) \leq 2\exp(-2D\epsilon^2).$$

- If $A$ and $B$ are generated randomly and conditioned on their Hamming distance being $dD$, then the mismatch positions mentioned above are uniformly random for every index;

□

To summarize, this proposition provides a new perspective to define the Hamming distance in the large hyperspace. With sufficiently large dimensionality $D$, Hamming distance between hypervectors can be interpreted statistically as the empirical mean of i.i.d. Bernoulli random variables, yielding a probabilistic characterization of similarity.

### A.3 THEOREMS

**Theorem 1** (Robust to Input Noise). *Let $s^{(1)}, s^{(2)} \in \mathcal{S} \subseteq \mathbb{R}^d$ be two feature vectors. $S = f_{\mathcal{ID}, \mathcal{L}, \Theta}(s)$ denotes the hypervector mapping defined in Equation 2. Suppose that for all indices of the features $n \in \{1, \ldots, d\}$,*

$$\frac{\left| x_n^{(1)} - x_n^{(2)} \right|}{\Delta_i} \leq \delta,$$

*where $\Delta_i$ is the difference between the upper and lower bounds of the $i^{th}$ feature value, and $\delta \in [0, 1]$ denotes the maximum distance ratio of the corresponding features. Then, with a sufficiently large $D$, the expected upper-bound Hamming distance between $S^{(1)} = f_{\mathcal{ID}, \mathcal{L}, \Theta}(s^{(1)})$ and $S^{(2)} = f_{\mathcal{ID}, \mathcal{L}, \Theta}(s^{(2)})$ converges to a monotonically increasing function $g$ with parameter $\delta$*

$$\mathbb{E}\left[ sup\left( d_H\left( S^{(1)}, S^{(2)} \right) \right) \right] \to g(\delta), \tag{22}$$

*This implies that the Hamming distance between a sample and its noisy variant is almost surely bounded by $g(\delta)$.*

*Proof.* We consider the $L^{(l_n(x_n^{(1)}))}, L^{(l_n(x_n^{(2)}))}$ first. Since $\frac{\left| x_i^{(1)} - x_i^{(2)} \right|}{\Delta_i} \leq \delta$, we can bound the difference of their corresponding Level Hypervectors. By definition 4, we have

$$
\begin{aligned}
& d_H\left( L^{(l_n(x_i^{(1)}))}, L^{(l_n(x_i^{(2)}))} \right) \\
& = \frac{l_n(x_i^{(1)}) - l_n(x_i^{(2)})}{M - 1} \\
& \leq \frac{1}{M - 1}\left( \frac{\delta \Delta_i}{\theta_{i, M-1} - \theta_{i, 1}}(M - 2) + 2 \right) \\
& \leq \frac{M}{M - 1}\delta + \frac{2}{M - 1} = \delta_d,
\end{aligned}
\tag{23}
$$

With the random flipping performed by $\otimes ID^{(i)}$, $\{L^{(l_n(x_i^{(1/2)}))} \otimes ID^{(i)}\}$ can be viewed as a new set of randomly generated hypervectors, denoted as $\{T^{i(1/2)}\}$, which satisfy the following relationship with Lemma 2

$$
\begin{aligned}
& d_H\left( T^{i(1)}, T^{i(2)} \right) \\
& = d_H\left( L^{(l_n(x_i^{(1)}))} \otimes ID^{(i)}, L^{(l_n(x_i^{(2)}))} \otimes ID^{(i)} \right) \\
& = d_H\left( L^{(l_n(x_i^{(1)}))}, L^{(l_n(x_i^{(2)}))} \right) \leq \delta_d.
\end{aligned}
\tag{24}
$$

To estimate the upper bound, we treat the inequality as an equality. Furthermore, with Proposition 1, we consider the Hamming distance constraint as a statistically condition, so that for every $T^{i(1)}$ and $T^{i(2)}$ pair, the equality situation at the $j^{th}$ index follows a Bernoulli distribution

$$\mathbb{I}\left( T_j^{i(1)} \neq T_j^{i(2)} \right) \sim_{i.i.d.} \text{Bernoulli}(\delta_d), \forall j \in \{1, 2, \ldots, D\}.$$

Next, we consider the situation on the $j^{th}$ index of $S^{(1)} = \bigoplus_{i=1}^d T^{i(1)}$ and $S^{(2)} = \bigoplus_{i=1}^d T^{i(2)}$ as a random variable $Z_j$ that satisfies

$$Z_j = \mathbb{I}\left( S_{(j)}^{(1)} \neq S_{(j)}^{(2)} \right)$$

Let $p_j^{(1/2)}$ be the number of $+1$ in the $\{T_j^{i(1/2)}\}_{i=\{1,2,\ldots,d\}}$, so that the probability of $Z_j = 1$ is

$$P(Z_j = 1) = \sum_{n=0}^{d} P(Z_j = 1 | p_j^{(1)} = n) \times P(p_j^{(1)} = n).$$

Specifically, by considering the $\{T_j^{i(1)}\}_{i=\{1,2,\dots,d\}}$ follow the i.i.d. Bernoulli(0.5), we have

$$P(p_j^{(1)} = n) = \frac{C_d^n}{2^d}.$$

Consider the situation where $n < \frac{d}{2}$ and $d$ is an odd number to avoid unnecessary complexity, we have the number of $+1$ in the second sample follow a combined Binomial distribution

$$p_j^{(2)}\big|_{p_j^{(1)}=n} \sim \text{Binomial}\,(n, 1 - \delta_d) + \text{Binomial}\,(d - n, \delta_d),$$

Therefore,

$$P(Z_j = 1 | p_j^{(1)} = n) = \sum_{k=\frac{d+1}{2}}^{d} \sum_{i=\max(0,\,k-d+n)}^{\min(n,\,k)} C_n^i \cdot C_{d-n}^{k-i} \cdot \delta^{n+k-2i}(1 - \delta)^{d-n-k+2i}.$$

For situation that $n > \frac{d}{2}$, we obtain an entirely analogous result

$$P(Z_j = 1 | p_j^{(1)} = n) = \sum_{k=0}^{\frac{d-1}{2}} \sum_{i=\max(0,\,k-d+n)}^{\min(n,\,k)} C_n^i \cdot C_{d-n}^{k-i} \cdot \delta^{n+k-2i}(1 - \delta)^{d-n-k+2i}.$$

With the derived results, we can calculate the probability of $P(Z_j = 1)$ as a function of $\delta_d$.

$$g(\delta) = p(\delta_d) = p(\frac{M}{M - 1}\delta + \frac{2}{M - 1}) \tag{25}$$

Finally, we consider the average with all the indices make the distribution arbitrarily close to the expectation, which is the probability we just calculated in Eqn. 25, thus complete the proof.

$$\square$$

This theorem establishes an approximate form of "continuity", showing that the designed mapping ensures that when the input noise is small, the distance between the corresponding mapped outputs is also bounded. In particular, this design exhibits better rejection properties against small perturbations with sufficiently large dimensionality $D$ and relatively large features $d$.

**Theorem 2** (Distance Between Cluster Prototype and Constituents). *Let $S^{(1)}, S^{(2)}, \dots, S^{(N)} \in \mathcal{H}^D$ be independently sampled random hypervectors. Define their sum as $C = [S^{(1)} + S^{(2)} + \dots + S^{(N)}]$. As $D \to \infty$, for any random hypervector $S^* \in \mathcal{H}^D$, index $j \in \{1, \dots, N\}$, the Hamming distance between $C$ and any component $S^{(n)}$ satisfies*

$$P\left(d_H(C, S^{(n)}) < d_H(C, S^*)\right) \to 1.$$

*Proof.* We begin with analyzing the probability that the $i^{th}$ index of $C$ and $S^{(n)}$ is different.

Let $Z_i^{(n)} = \sum_{k \neq n} S_i^{(k)}$, where $S_i^{(k)}$ is the $i^{th}$ value of $S^{(k)}$, so that:

$$C_i = \left[\sum_{k=1}^{N} S_i^{(k)}\right] = \left[S_i^{(n)} + \sum_{k \neq n} S_i^{(k)}\right] = \left[S_i^{(n)} + Z_i^{(n)}\right].$$

Since the random hypervectors are independently sampled, the $N - 1$ terms $S_i^{(k)}$ $(k \neq n)$ are independent random variables satisfying:

$$\mathbb{P}(S_i^{(k)} = +1) = \mathbb{P}(S_i^{(k)} = -1) = 0.5.$$

Since $\frac{S_i^{(k)}+1}{2} \sim \text{Bernoulli}(0.5)$, their i.i.d. sum $Z_i^{(n)}$ follows a shifted binomial distribution:

$$Z_i^{(n)} \sim 2 * \text{Binomial}(N-1, 0.5) - (N-1)$$

Given $S_i^{(n)} = +1$, we have $C_i = [1 + Z_i^{(n)}]$. Without losing generality, we only consider the situation when $N-1$ is an even number, which lead to

$$P(C_i \neq S_i^{(k)}) = P([1 + Z_i^{(n)}] \neq 1) = P(Z_i^{(n)} < -1) + \frac{1}{2}P(Z_i^{(n)} = -1) = P(Z_i^{(n)} < -1),$$

where $P(Z_i^{(n)} = -1) = 0$ since the sum can only take even values.

With the distribution of $Z_i^{(n)}$, we can further calculate the specific probability:

$$P(C_i \neq S_i^{(n)}) = \frac{1}{2} - 2^{-N} C_{N-1}^{\frac{N-1}{2}}.$$

By symmetry, the same calculation applies if $S_i^{(n)} = -1$. Notice that $\mathbb{I}(C_i \neq S_i^{(n)})$ presents a Bernoulli distribution and this process is independently carried out for every index, we can derive the distribution of the Hamming distance $d_H(C, S^{(n)})$:

$$d_H(C, S^{(n)}) = \frac{1}{D} \sum_{i=1}^{D} \mathbb{I}(C_i \neq S_i^{(n)}) \sim \frac{1}{D}\text{Binomial}(D, \frac{1}{2} - p(N))$$

where $p(N) = 2^{-N} C_{N-1}^{\frac{N-1}{2}}$.

Due to independently sampling, we can view $S^*$ and $B$ just as two random hypervectors in the hyperspace, so we can apply the result from Lemma 3, we know that

$$d_H(C, S^*) \sim \frac{1}{D}\text{Binomial}(D, \frac{1}{2}).$$

With central limit theorem, these two distributions converge to normal distributions as $D \to \infty$

$$d_H(C, S^*) \sim \mathcal{N}(\frac{1}{2}, \frac{1}{4}D^{-1}), \quad d_H(C, S^{(n)} \sim \mathcal{N}(\frac{1}{2} - p(N), (\frac{1}{4} - p^2(N))D^{-1}).$$

Based on that, we have,

$$\begin{aligned}
&P\left(d_H(C, S^{(n)}) - d_H(C, S^*) \geq 0\right) \\
&= P\left((d_H(C, S^{(n)}) - \frac{1}{2} - \frac{p(N)}{2}) - (d_H(C, S) - \frac{1}{2} - \frac{p(N)}{2}) \geq 0\right) \\
&\leq P\left((d_H(C, S^{(n)}) - \frac{1}{2} - \frac{p(N)}{2}) \geq 0\right) + P\left((d_H(C, S) - \frac{1}{2} - \frac{p(N)}{2}) \leq 0\right) \quad (26)\\
&\leq P\left(\left|d_H(C, S^{(n)}) - \frac{1}{2} - p(N)\right| \geq \frac{p(N)}{2})\right) + P\left(\left|d_H(C, S) - \frac{1}{2}\right| \geq \frac{p(N)}{2}\right) \\
&= 2 * \Phi(-p(N)D^{\frac{1}{2}}) + 2 * \Phi(-\frac{p(N)}{\sqrt{1 - 4p^2(N)}}D^{\frac{1}{2}})
\end{aligned}$$

we can select appropriate $D = D(p(N))$ to allow the right-hand side of the equation to approach zero arbitrarily by letting $p(N)D^{\frac{1}{2}} \to \infty$. Thus we have the result

$$P\left(d_H(C, S^{(n)}) < d_H(C, S^*)\right) \to 1,$$

where $\Phi$ is the Cumulative Distribution Function of the standard normal distribution.

Moreover, we further know that the convergence is roughly characterized $\Phi(-p(N)D^{\frac{1}{2}})$.

Notice that $p(N) \to 0$ as $N \to \infty$, and the dimension $D$ is required to be dependent on $p(N)$ to make the scale become effective, we apply $N >> D$ for the implementation. Intuitively speaking, we require a sufficiently large $N$ to divide this two normal peak apart given a small $p(N)$.

$\square$

**Theorem 3** (Robustness to Hardware Error). *Assume we have a sample hypervector $S$ and two cluster hypervectors $C_1$ and $C_2$, whose initial Hamming distances satisfy:*

$$d_H(S, C_1) = d_1 \quad and \quad d_H(S, C_2) = d_2,$$

*where $d_2 - d_1 = \epsilon > 0$.*

*We randomly flip a proportion $p$ ($p < 0.5$) of the bits in both $C_1$ and $C_2$, yielding two new hypervectors $C_1'$ and $C_2'$. As $D \to \infty$, we have*

$$P\big(d_H(S, C_1') < d_H(S, C_2')\big) \to 1$$

*Proof.* Again, we begin with the situation at the $i^{th}$ index. Let $X_i$ and $Y_i$ be the indicator variables denoting whether the $i$-th bit differs from $S$ after corruption:

$$X_i = \mathbf{1}_{\{(C_1')_i \neq S_i\}}, \qquad Y_i = \mathbf{1}_{\{(C_2')_i \neq S_i\}},$$

so that the post-corruption Hamming distances are:

$$d_H(S, C_1') = \frac{1}{D} \sum_{i=1}^{D} X_i, \qquad d_H(S, C_2') = \frac{1}{D} \sum_{i=1}^{D} Y_i.$$

Next, we analyze the expectation of each $X_i$ and $Y_i$. With the view of Proposition 1, with a sufficiently large $D$, we can transfer the Hamming distance $d$ as a random event with probability $d$:

- For each $i$, if $C_1$ and $S$ originally differ at bit $i$, which happens with probability $d_1$, then flipping that bit with probability $p$ yields:
$$\mathbb{P}(X_i = 1 \mid \text{originally different}) \to 1 - p, \qquad \mathbb{P}(X_i = 0 \mid \text{originally different}) \to p.$$

- If they originally agree (probability $1 - d_1$), then:
$$\mathbb{P}(X_i = 1 \mid \text{originally same}) \to p, \qquad \mathbb{P}(X_i = 0 \mid \text{originally same}) \to 1 - p.$$

Hence, the expectation becomes:
$$\mathbb{E}[X_i] \to d_1(1 - p) + (1 - d_1)p = d_1(1 - 2p) + p.$$

Similarly,
$$\mathbb{E}[Y_i] = d_2(1 - 2p) + p.$$

Since $d_2 = d_1 + \epsilon$ and $1 - 2p > 0$, we have:
$$\mathbb{E}[Y_i] - \mathbb{E}[X_i] = (d_2 - d_1)(1 - 2p) = \epsilon(1 - 2p) > 0.$$

Now define the total difference in post-corruption Hamming distances:
$$Z = \sum_{i=1}^{D} (Y_i - X_i).$$

Then:
$$\mathbb{E}[Z] = D \cdot (\mathbb{E}[Y_i] - \mathbb{E}[X_i]) = D \cdot \epsilon(1 - 2p).$$

Because $X_i, Y_i$ are bounded, independent random variables, and the variance of each term is bounded, we can apply Hoeffding's inequality to show that:
$$\mathbb{P}(Z < 0) \leq \exp(-cD)$$

for some constant $c > 0$. This implies:
$$\mathbb{P}(d_H(S, C_1') < d_H(S, C_2')) = \mathbb{P}(Z > 0) \to 1 \quad \text{as } D \to \infty.$$

$\square$

