# OpenReview forum: "HDC-X: Efficient Medical Data Classification for Embedded Devices"
_ICLR.cc/2026/Conference — ICLR 2026 Conference Withdrawn Submission_

### Official Review · Reviewer_VuG2 · 2025-10-21

**Soundness:** 2
**Presentation:** 2
**Contribution:** 2
**Rating:** 4
**Confidence:** 2

**Summary:**

This paper introduces HDC-X, a lightweight and energy-efficient framework for medical data classification on low-power embedded devices. Built upon hyperdimensional computing principles, HDC-X encodes input features into high-dimensional hypervectors, aggregates them into cluster-specific prototypes, and performs classification through similarity search in hyperspace. The method offers a strong balance between accuracy, robustness, and computational efficiency.

Across three medical tasks—including heart sound, breast cancer, and electromyography classification—HDC-X achieves up to 350× higher energy efficiency than state-of-the-art deep learning models such as Bayesian ResNet, with less than 1% accuracy difference. The paper also provides theoretical proofs of robustness against input noise and hardware errors, and presents a conceptual hardware framework demonstrating feasibility for FPGA implementation. Together, these contributions make HDC-X a promising step toward practical, reliable AI-based disease screening on portable and resource-constrained medical devices.

**Strengths:**

This paper presents an original and well-executed contribution to energy-efficient medical data classification by extending the hyperdimensional computing (HDC) paradigm into a practical, high-performance framework, HDC-X. The approach is both conceptually creative and technically sound: it reformulates medical classification as a similarity search in high-dimensional hyperspace using rigorously defined Hamming-distance–based metrics. This yields a framework that preserves the lightweight, brain-inspired nature of HDC while substantially improving accuracy and robustness.

The theoretical formulation is rigorous, with clear proofs and precise definitions that make the methodology transparent and reproducible. The empirical validation is comprehensive—covering multiple datasets, hardware efficiency analysis, and robustness under input noise, limited data, and hardware faults—demonstrating impressive real-world relevance.

**Weaknesses:**

While the paper demonstrates strong performance on small- to medium-scale medical datasets, it remains unclear whether the proposed HDC-X framework can generalize effectively to larger-scale settings. The current empirical evaluation does not explore the scaling behavior of HDC-X—such as how its accuracy, energy efficiency, or robustness evolve with increasing data size or feature dimensionality. Without such analysis, it is difficult to assess whether the observed advantages over deep neural networks persist when applied to more complex or higher-volume medical data. Future work could strengthen the contribution by providing experiments or theoretical insights into the scalability of HDC-based methods.

**Questions:**

This is an interesting paper, though the empirical results are somewhat limited. It is important to further investigate the scaling properties of hyperdimensional computing (HDC) to substantiate its claimed energy efficiency. Currently, the experiments are conducted on relatively small datasets, where careful hyperparameter tuning may mask potential weaknesses—particularly the tendency of larger models to overfit or lose efficiency at scale. A more thorough analysis on larger and more complex datasets would help confirm whether HDC maintains its advantage in energy efficiency under realistic, large-scale conditions.

---

### Official Review · Reviewer_1EHd · 2025-10-30

**Soundness:** 2
**Presentation:** 2
**Contribution:** 1
**Rating:** 0
**Confidence:** 4

**Summary:**

This paper proposes HDC-X, a lightweight and energy-efficient classification framework tailored for medical data analysis on low-power and embedded devices. Building upon Hyperdimensional Computing (HDC) principles, HDC-X encodes medical signals into high-dimensional hypervectors, aggregates them into cluster-specific prototypes (Cluster-HVs), and performs classification via similarity search in hyperspace.

**Strengths:**

By leveraging hyperdimensional computing (HDC), HDC-X eliminates the need for floating-point operations and GPUs, enabling extremely low-power, memory-efficient inference

**Weaknesses:**

1. The proposed method primarily reuses the standard HDC pipeline (encoding → bundling → similarity search), with only moderate extension via cluster aggregation. There is no clear conceptual innovation beyond applying HDC to biomedical data, which makes the contribution more application-level than methodological.
2. The paper does not fundamentally address the long-standing issue that HDC-based models often underperform deep neural networks in complex, high-dimensional feature spaces. While the reported 1% gap to Bayesian ResNet is notable, it is unclear whether this holds across more challenging datasets or higher intra-class variability.
3. In practical HDC implementations, the encoding stage (mapping raw features into hypervectors via item memory or continuous item memory) dominates both latency and energy consumption. However, the paper omits reporting encoding cost or how the chosen encoder scales in time, memory, and energy. This is a critical omission because the efficiency advantage may vanish once encoding overhead is included.
4. The paper fixes the hypervector dimension and precision but does not explore how these parameters affect accuracy, energy, or latency. Such analysis is crucial, as HDC accuracy is highly sensitive to dimensionality and quantization (binary vs. int8 vs. float32). Without this exploration, the claimed “energy efficiency” lacks reproducibility and generalization.
5. The reported 350× efficiency improvement appears to be a theoretical estimate rather than measured on actual hardware (e.g., MCU, FPGA, or ASIC). Without concrete energy profiling of encoding, similarity computation, and memory access, the practical significance of this number is uncertain.
6. The experiments are limited to a small set of biomedical datasets (e.g., heart sound classification). Comparisons exclude other lightweight neural or neuromorphic baselines such as quantized CNNs, TinyML architectures, or SNN-based classifiers, which would provide a fairer context for energy–accuracy trade-offs.

**Questions:**

see weakness

---

### Official Review · Reviewer_jWQS · 2025-10-30

**Soundness:** 2
**Presentation:** 2
**Contribution:** 1
**Rating:** 2
**Confidence:** 5

**Summary:**

This paper presents HDC-X, which leverages hyperdimensional computing to enable lightweight medical data classification. The proposed methods do class-wise clustering, which does bundling and optional retraining, after encoding (mapping data to high-dimensional space). The authors claim that HDC-X can achieve similar accuracy to a deep-learning-based method while being up to 350x more energy-efficient for classifying medical datasets.

**Strengths:**

The propsed method significantly improves energy efficiency and noise resiliency compared to deep neural networks while maintaining comparable accuracy.

**Weaknesses:**

This paper shows that it can achieve similar accuracy with much less cost compared to a deep learning-based method. However, there are several points that need to be addressed:
1. The huge improvements and robustness of HDC-X come from HDC's strength. The encoding, training (bundling), and retraining methods that are presented in this paper are not new, which are from VoiceHD (Imani 2017) and HDCluster (Imani 2019), etc, so I am not very convinced about the novelty of this work.
2. The target task can be very easy to achieve with small and lightweight DNNs, but the baseline DNNs seem to be very heavy. Also, at the inference time, the DNN can be compressed with lower precision for better efficiency.
3. Baseline implementation details, such as HDC, HDCluster, etc, are not very clear, and I think the authors should present HDC-X's difference over baselines, especially HDC-based ones.
4. Theoretical proof seems to be already done by [1], and I would like to know theoretical contribution of this paper.
[1] Anthony Thomas et al. A Theoretical Perspective on Hyperdimensional Computing. JAIR, 2021.

**Questions:**

Please see weakness section.

---

### Official Review · Reviewer_cWEf · 2025-11-02

**Soundness:** 3
**Presentation:** 3
**Contribution:** 2
**Rating:** 2
**Confidence:** 5

**Summary:**

HDC-X extends hyperdimensional computing for energy-efficient medical classification by introducing cluster-based prototype hypervectors for each class. The method targets embedded devices where power and latency are critical. Results show strong energy savings relative to deep learning models, and the paper discusses a potential FPGA realization.

**Strengths:**

The paper contributes an applied demonstration of HDC’s advantages for low-power medical tasks. Although the motivation is compelling, the methodological advancement is modest, and the lack of concrete hardware results limits its impact.
• The energy-efficient focus is timely and important, addressing the need for lightweight edge solutions in healthcare.
• Reported improvements over heavy neural baselines are large and highlight potential for practical deployment.
• The approach reinforces the viability of HDC models in real-time embedded environments.

**Weaknesses:**

• The algorithmic innovation beyond prior HDC clustering methods is minimal and not clearly isolated.
• Comparisons omit lighter classical baselines, making the energy-efficiency claims somewhat inflated.
• The FPGA analysis remains conceptual, with no real measurements or synthesis details.

**Questions:**

Could the authors provide actual measured energy or latency results from hardware deployment? How does HDC-X compare to simple non-neural baselines on the same datasets? What specific architectural change differentiates HDC-X from previous prototype-HDC models?

---

### Note · Authors · 2025-11-19

**Comment:**

After discussion, we have decided to withdraw our paper. As ICLR requires all reviews, including rejected and withdrawn ones, to be released on OpenReview, we have chosen to include the following statement for future readers.

---

We thank the reviewers for their time. However, upon reading the reviews, it became apparent that one or more reviewers have not read key sections of the paper, if not the entire manuscript. Several claims in the reviews are directly contradicted by the contents of our submitted paper. Some specific examples are:


**Claim:** (Reviewer 1EHd) The reported 350x efficiency gain is theoretical, not measured on hardware.

**Fact:** All energy results were empirically measured on physical hardware using a 10-fold evaluation (Table 1). Section 4.2 is dedicated to showing how energy is measured and what hardware is used.


**Claim:** (Reviewer 1EHd) The effect of hypervector dimensionality on accuracy is not explored.

**Fact:** Figure 4(a) is dedicated to this.


**Claim:** (Reviewer 1EHd) Hypervector encoding cost is omitted, potentially eliminating the efficiency advantage.

**Fact:** All energy measurements includes encoding energy.


**Claim:** (Reviewer cWEf, 1EHd) The evaluation lacks comparisons against non-neural or lightweight neural baselines.

**Fact:** Table 1 includes comparisons against K-Means, Standard HDC, HDCluster, Light CNN and Pruned Neural Networks.

---

We also recognize that certain aspects of our work may not have been emphasized sufficiently in the paper, and we consider these our responsibility. We hope to address them here:


**Claim:** (Reviewer cWEf, jWQS) HDC-X is essentially HDC/HDCluster and has already been presented in prior work.

**Fact:** We compare HDC-X with against HDC and HDCluster in Table 1. HDC-X achieves ~10% higher accuracy than either method on heart sound classification. Notably, HDC-X is the first to introduce in-class hyperspace clustering for hypervector prototypes, allowing the model to better capture intra-class variability and improve robustness. We acknowledge that we did not explicitly isolate and highlight these architectural differences from prior HDC-based approaches, and will make them explicit in our next submission.


**Claim:** (Reviewer 1EHd, VuG2) Experiments use only easy datasets and do not address HDC underperformance on challenging tasks.

**Fact:** This work specifically aims to improve the performance of low-cost HDC models on high-variability, more complex datasets. HDC-X is evaluated on three major medical benchmarks, including PhysioNet 2016 — the largest and most challenging heart sound dataset to date (*Potes et al.* reported only 86.02% accuracy as the challenge-winning model). HDC-X improves accuracy from 77.84% (Standard HDC) to 88.18%, placing it within 1% of the extremely heavy Bayesian ResNet. We also acknowledge that HDC-X is not intended for ImageNet-level scale or complexity, as it remains a non-neural, one-shot model.


**Claim:** (Reviewer jWQS) The theoretical proof seems to have already been done by *Thomas et al.*

**Fact:** We are aware of this paper and have cited it in our manuscript. *Thomas et al.* established results in *continuous* hyperspace, whereas our work provides theoretical guarantees for *binary* hypervectors, which are more widely used in modern HDC systems and, to the best of our knowledge, have not been formally proven in prior HDC research.

---

We want to emphasize that our concerns do not apply to all comments. Some reviewers clearly read the manuscript and provided thoughtful, constructive feedback. We appreciate those comments, including the critical ones, and we will incorporate them to strengthen the next submission of this work. We sincerely thank those reviewers for their effort, fairness, and engagement.

Unfortunately, some of the errors are so fundamental that they cannot be reasonably explained by time pressure or minor oversight. Several patterns strongly suggest that at least one reviewer may have relied heavily or entirely on AI tools rather than fulfilling their fundamental responsibility of personally reading and evaluating the manuscript.

This is unacceptable, especially at a conference of such prestige.

We want to remind these reviewers that the first instruction in the [ICLR Reviewer Guide](https://iclr.cc/Conferences/2026/ReviewerGuide?utm_source=chatgpt.com#:~:text=Reviewing%20a%20submission:%20step%2Dby%2Dstep) is “Read the paper”. Although it should be unnecessary to repeat this to experienced researchers, we quote:

> “1. Read the paper: It’s important to carefully read through the entire paper and to look up any related work and citations that will help you comprehensively evaluate it. Be sure to give yourself sufficient time for this step.”

We have decided to withdraw this paper because we do not believe the reviews, taken as a whole, provide a fair or accurate evaluation of the work. We include this statement so that future readers understand that many of the issues described as “omit” or “unexplored” in the reviews are, in fact, present in the manuscript. We encourage readers to consult the original paper rather than rely solely on the reviews when forming their judgment.

**Withdrawal Confirmation:**

I have read and agree with the venue's withdrawal policy on behalf of myself and my co-authors.